

# Photocatalysis and adsorption kinetics of azo dyes by nanoparticles of nickel oxide and copper oxide and their nanocomposite in an aqueous medium

Hajra Ahsan[1], Muhammad Shahid[2], Muhammad Imran[3], Faisal Mahmood[1], Muhammad Hussnain Siddique[2], Hayssam M. Ali[4], Muhammad B.K. Niazi[5], Sabir Hussain[1], Muhammad Shahbaz[6], Mudassar Ayyub[1] and Tanvir Shahzad[1]

[1] Department of Environmental Sciences, Government College University, Faisalabad, Pakistan
[2] Department of Bioinformatics and Biotechnology, Government College University, Faisalabad, Pakistan
[3] Department of Environmental Sciences, COMSATS University Islamabad, Vehari-Campus, Vehari, Pakistan
[4] Botany and Microbiology Department, College of Science, King Saud University, Riyadh, Saudi Arabia
[5] School of Chemical and Materials Engineering, National University of Sciences & Technology, Islamabad, Pakistan
[6] Centre for Environmental and Climate Science, Lund University, Lund, Sweden

Corresponding author
Tanvir Shahzad,
hereistanvir@gmail.com

## ABSTRACT

**Background.** Azo dyes are recalcitrant organic pollutants present in textile industry effluents. Conventional treatment methods to remove them come with a range of disadvantages. Nanoparticles and their nanocomposites offer more efficient, less expensive and easy to handle wastewater treatment alternative.

**Methods.** In this study, nanoparticles of nickel oxide (NiO-NPs), copper oxide (CuO-NPs) and their nanocomposite (NiO/CuO-NC) were synthesized using co-precipitation method. The functional groups present on the surface of synthesized nanomaterials were verified using Fourier-transform infrared spectroscopy (FTIR). Surface morphology was assessed using scanning electron microscopy (SEM) whereas purity, shape and size of the crystallite were determined using X-ray diffraction (XRD) technique. The potential of these nanomaterials to degrade three dyes *i.e.*, Reactive Red-2 (RR-2), Reactive Black-5 (RB-5) and Orange II sodium salt (OII) azo dyes, was determined in an aqueous medium under visible light (photocatalysis). The photodegradation effectiveness of all nanomaterials was evaluated under different factors like nanomaterial dose (0.02–0.1 g 10 mL$^{-1}$), concentration of dyes (20–100 mg L$^{-1}$), and irradiation time (60–120 min). They were also assessed for their potential to adsorb RR-2 and OII dyes.

**Results.** Results revealed that at optimum concentration (60 mgL$^{-1}$) of RR-2, RB-5, and OII dyes, NiO-NPs degraded 90, 82 and 83%, CuO-NPs degraded 49, 34, and 44%, whereas the nanocomposite NiO/CuO-NC degraded 92, 93, and 96% of the said dyes respectively. The nanomaterials were categorized as the efficient degraders of the dyes in the order: NiO/CuO-NC > NiO-NPs > CuO-NPs. The highest degradation potential shown by the nanocomposite was attributed to its large surface area, small particles size, and quick reactions which were proved by advance analytical techniques. The equilibrium and kinetic adsorption of RR-2 and OII on NiO-NPs, CuO-NPs, and NiO/CuO-NC were well explained with Langmuir and Pseudo second order model,

respectively ($R^2 \geq 0.96$). The maximum RR-2 adsorption (103 mg/g) was obtained with NiO/CuO-NC. It is concluded that nanocomposites are more efficient and promising for the dyes degradation from industrial wastewater as compared with dyes adsorption onto individual NPs. Thus, the nanocomposite NiO/CuO-NC can be an excellent candidate for photodegradation as well as the adsorption of the dyes in aqueous media.

## INTRODUCTION

Release of untreated or semi-treated wastewater from industries is almost a norm in developing countries due to less stringent environmental laws as well as their half-hearted enforcement (*Imran et al., 2019*; *Iqbal et al., 2019*). Textile industry is one of the major industries releasing about 50% of the applied azo dyes in wastewater, which adversely affect the environment, humans, and plants even when present in trace amount (*Malakootian et al., 2015*; *Rehman et al., 2018*; *Yaseen & Scholz, 2019*; *Shemawar et al., 2021*). The elevated solubility and complicated structure of synthetic dyes make them persistent and difficult for degradation in nature. Several conventional techniques have been extensively employed for their removal. However, they have limitations like high cost, short application life, and incomplete removal (*Kumari, 2017*; *Cai et al., 2020*; *Marrakchi, Hameed & Hummadi, 2020*). Photocatalytic degradation (*Cai et al., 2022*; *Li et al., 2022*) as an advanced oxidation technique has attained significant attention owing to its excellent potential of environmental remediation, efficient degradation ability, easy to use, reliable, and less expensive technique (*Védrine, 2019*). In this context, nanomaterials have been found to be very effective photocatalytic agents and adsorbents for wastewater treatment due to their nanoscale size, tailored shape, reactivity as well as magnetic, electrical, and optical properties (*Khin et al., 2012*; *Abbas et al., 2022*; *Idrees et al., 2022*).

Nickel oxide nanoparticles (NiO-NPs) could be effective for dye degradation as a p-type semiconductor owing to their high chemical-stability, photo-stability, high transparency, and degradation efficiency (*Hafdi et al., 2020*). However, dye degradation efficiency is not much significant in response to single semiconductors due to the high recombination rate of electronic-hole pairs (*Muhambihai, Rama & Subramaniam, 2020*). The preparation and use of nanocomposites based on combined semiconductors is a promising method to avoid recombination rate of e/h pairs (*Khan et al., 2014*; *Yanyan et al., 2017*). Hence, various modifications have been applied to the nanoparticles using different metals, metal oxides, polymers, graphene, carbon nanotubes (CNT), and other carbonaceous materials to rationally enhance the morphological structure and photocatalytic characteristics of NiO-NPs (*Chen et al., 2020*; *Karimi-Maleh et al., 2020*).

Copper oxide nanoparticles (CuO-NPs) are readily available and commonly used p type semiconductors in modern technology due to their less toxicity, and high chemical and electrochemical stability (*Guajardo-Pacheco et al., 2010*; *Kannan et al., 2022*). According to

literature, such potential characteristics of CuO-NPs can prove effective when integrated with NiO-NPs for dye degradation (*Singh et al., 2016*). It can be used as support material, catalyst, and as effective adsorbent material for the degradation of organic effluents and sequestration of inorganic contaminants (*Idrees et al., 2021*). The photocatalytic degradation studies of dyes are based on applications of n-p, n-n, and p-p isotype heterojunction, leading to improvement in photocatalytic dye degradation. For instance, nanoparticles of Cerium (Ce), Iron (Fe), and $TiO_2$ were shown to photodegrade RR-2, O-II, and RB-5 dyes up to 78.8%, 71.50%, and 58%, under 80, 360, and 120 min UV irradiation (253.7 nm), respectively (*Lucas et al., 2007*; *Xu et al., 2012*; *Ezzatahmadi et al., 2018*). In another study, photocatalytic potential of NiO-NPs to degrade Congo red (CR), Methyl blue (MB), and Rhodamine B (Rh B), was found 84%, 98.7%, and 80.33%, respectively (*Khairnar & Shrivastava, 2019*; *Bhat et al., 2020*). Moreover, *Zaman et al. (2012)* reported the highest degradation (72%) of MB dye using chemically synthesized CuO-NPs in 24 h. On the other hand, composites have been found to lead to higher photocatalytic performance and adsorption potential than NPs, due to their high charge transfer properties, stability, strong interconnectivity, and degradation ability (*Ahmad et al., 2020*).

The present study was designed on the hypothesis that application of p-p isotype semiconducting nanocomposite (NiO/CuO-NC) would show higher photocatalytic degradation and adsorption potential for the removal of azo dyes than their constituent NPs. The NiO-NPs, CuO-NPs and NiO/CuO-NC were applied to explore their potential as a catalyst and adsorbent to remove three azo dyes (RB-5, RR-2, and OII) from aqueous solutions. These dyes have anionic structural properties of sodium salt which help to optimize the chemical reactions and to analyse the degradation efficiency of nanomaterials. The photodegradation and adsorption efficiency of synthesized nanomaterials was optimized through various parameters such as initial dye concentration, catalyst dose, and contact time. The adsorption experimental data for the dyes sequestration onto NiO-NPs, CuO-NPs and NiO/CuO-NC were evaluated through kinetic and equilibrium sorption isotherm models. The synthesized materials (NiO-NPs, CuO-NPs and NiO/CuO-NC) were characterized using FTIR, XRD, and SEM to elucidate the morphological and structural properties. Moreover, adsorption potential of as-synthesized materials for the removal of RR-2 and OII has been compared with the existing literature to find economic feasibility and recommendation for the dyes sequestration.

## MATERIALS & METHODS

### Chemicals

The analytical grade salts of nickel (II) chloride hexahydrate ($NiCl_2·6H_2O$), copper sulfate pentahydrate ($CuSO_4·5H_2O$), and sodium hydroxide (NaOH) were procured from a company (Sigma-Aldrich) and were used for experiment without further purification. Azo dyes *e.g.*, Reactive Red-2 (RR-2), Reactive Black-5 (RB-5), and Orange II sodium salt (OII) used for preparing the synthetic wastewater were also procured in their pure form.

## Synthesis of NiO, CuO nanoparticles and NiO/CuO nanocomposite

Nickle oxide nanoparticles (NiO-NPs) were prepared through a simple co-precipitation method. For this purpose, 500 mL of 0.1M $NiCl_2 \cdot 6H_2O$ was prepared in distilled water as a precursor of nickel oxide (NiO). The solution was magnetically stirred at 600 rpm using magnetic stirrer (PHOENIX Instrument Magnetic Stirrer RSM-01) over 50 °C for 30 min. Sodium hydroxide (0.5 M) was then added dropwise to increase the pH of the solution until a light green colour appeared (pH 8). The resulting precipitates in beaker were washed repeatedly using distilled water (DW) and ethanol. The content were subsequently filtered and the filtrate was collected in a beaker. The solution was then oven dried for 24 h at 60 °C. The dried NiO-NPs were ground with pestle and mortar and were separated into fine powder with a sieve (*Ai & Zeng, 2013*).

Same method was followed to synthesize the CuO-NPs. In this case, 500 ml of 0.1M solution $CuSO_4 \cdot 5H_2O$, as a precursor of copper oxide (CuO) was prepared in distilled water. The solution was magnetically stirred at 600 rpm using magnetic stirrer (PHOENIX Instrument Magnetic Stirrer RSM-01) for 30 min at 50 °C. After that, 0.5 M solution of NaOH was added dropwise to maintain the pH at 7 or until the appearance of black colour, an indication of the CuO-NPs synthesis. The resulting black precipitates in beaker were washed repeatedly using distilled water (DW) and ethanol followed by filtration. The pure solution was then oven dried for 24 h at 60 °C. The dried CuO-NPs were were ground to fine powder with a pestle and mortar (*Zaman et al., 2012*).

For the synthesis of NiO/CuO-NC, 12.48 g of $CuSO_4.5H_2O$ and 5.94 g of $NiCl_2 \cdot 6H_2O$ (1:0.5) were dissolved in 50 mL distilled water and the suspension was magnetically stirred at 600 rpm using magnetic stirrer (PHOENIX Instrument Magnetic Stirrer RSM-01) at 50 °C for 30 min. Afterwards, 0.5 M NaOH was added dropwise into the solution till the appearance of green blue colour *i.e.*, pH 11. In the final step, the obtained green blue gel was washed using distilled water (DW) and ethanol, then filtered with filter paper and filtrate was collected in a beaker and particles were oven dried at 60 °C for 24 h. The dried CuO/NiO-NC nanocomposite were ground to fine powder with pestle and mortar (*Aazam, 2014*).

## Material characterization

Scanning electron microscopy (SEM) analysis was performed using scanning electron microscope (TM-1000, Hitachi, Japan).

X-ray diffraction (XRD) patterns of the NiO-NPs, CuO-NPs and NiO/CuO-NC were determined through X-ray diffractometer (STOE-Germany) with monochromator working at Cu K $\alpha$ $\lambda = 0.15418$ nm, voltage = 45 kV, and current = 40 mA. The samples (nanomaterials) were recorded at a step size = 0.04, step time = 0.5s/step and $2\theta$ ranged from 20°–60°. The PowderCell software was used to analyse the phase purity and crystal structure of nanomaterials. Debye–Scherer's formula (Eq. (1)) was employed to calculate the average particle size of nanomaterials (*Kesavamoorthi et al., 2016*).

$$D = \frac{k\lambda}{\beta\cos\theta} \tag{1}$$

where, 'D' is the crystallite size; 'k' is the shape factor (0.94); '$\beta$' is the Full Width at Half Maximum (FWHM); '$\theta$' is the diffraction angle; '$\lambda$' is the wavelength of X-ray (0.1546 nm).

The Fourier-transform infrared (FTIR) spectral analysis of the samples was carried out using Attenuated Total Reflectance Fourier Transform Infrared Spectrometer (ATRFTIR, BRUKER, USA) in range of 4,000 cm$^{-1}$ to 300 cm$^{-1}$ wave number, at resolution of 4 cm$^{-1}$ and scanning frequency of 32. The spectra of samples were determined in KBr pellets which were dried at 100 °C overnight.

## Dyes degradation experiment anddata analysis
### Preparation of synthetic dyes
Stock solutions (5,000 mg L$^{-1}$) of Reactive Red-2, Reactive Black-5, and Orange II sodium salt dyes were prepared using 0.25 g of each dye in 50 mL distilled water separately and working solutions of each dye were prepared by diluting the respective stock solution.

## Photocatalytic and adsorptive sequestration of dyes
The experiments for the photocatalytic removal of RB-5, RR-2, and OII dyes and adsorptive removal of RR-2, and OII dyes were performed using the as-synthesized nanomaterials (NiO-NPs, CuO-NPs, and NiO/CuO-NC). The tungsten lamps (420 nm wavelength) in photoreactor were used to perform the photocatalysis reaction. The experimental data of both studies were measured spectrophotometrically (Stalwart STA−82000 V) after 2 h' irradiation over visible light and dark reaction using spectrophotometer cuvettes (1202K83, Optical Polystyrene, 1180W04, cuvette Methacrylate, 4.5 mL clear sides) respectively. The photodegradation study was conducted by varying the catalysts doses *i.e.*, 0.02, 0.04, 0.06, 0.08, and 0.1 g 10 mL$^{-1}$, and dye concentrations *i.e.*, 20, 40, 60, 80 and 100 mg L$^{-1}$ to study the impact of various parameters on dyes removal from contaminated water.

For adsorptive study, 250 mL solution of dyes (60 mg L$^{-1}$) was prepared using 0.15 g of all nanomaterials under continuous shaking at a constant pH (7.5). Then, 0.5 M solution of NaOH was used to maintain pH of dyes solution using a pH meter (Milwaukee pH 55, Thailand). Various concentration of dyes (40–100 mg L$^{-1}$) were prepared and aliquots of 1.5 mL were taken after different time intervals (15, 30, 60, 120 min) to determine the equilibrium and kinetic adsorption and their simulation with isotherm and kinetics models. Absorbance of RR-2, RB-5, and OII dyes was recorded at 540 nm, 597 nm, and 485 nm, respectively using UV–vis spectrophotometer (Stalwart STA−82000 V).

The photodegradation (removal %) and adsorption capacity of dyes was calculated using Eqs. (2) and (3), respectively (*Shah et al., 2018*; *Imran et al., 2019*).

$$\text{Degradation or removal (\%)} = \left( \frac{C_0 - C_f}{C_0} \right) \times 100 \tag{2}$$

$$\text{Adsorption (mg/g)} = \left( \frac{C_0 - C_f}{M_a} \right) \times V_w \tag{3}$$

where $C_0$ and $C_f$ are the initial and final concentrations of each dye (mg L$^{-1}$) after interaction with the materials. In Eq. (3), $V_w$ is the volume of dye containing water (L) and $M_a$ is the mass of the nonmaterial (g) used for experimentation.

## Adsorption isotherms

The adsorption isotherm shows the equilibrium relationship between active sites on the adsorbent surface and the contaminant (*Akram et al., 2019*; *Din et al., 2021*; *Imran et al., 2021*). The experimental equilibrium adsorption data for the removal of RR-2 and OII by NiO-NPs, CuO-NPs and NiO/CuO-NC were simulated with Langmuir (Eq. (4)) and Freundlich (Eq. (6)) models. The concentration of dyes was varied from 40–100 mg L$^{-1}$ while other parameters *i.e.*, time interval (2 h), nanomaterial dosage (0.06 g 10 mL$^{-1}$), and pH (7.5) were kept constant (*Imran et al., 2019*; *Iqbal et al., 2019*; *Tran et al., 2017*).

$$q_e = \left[ \frac{K_L C_e Q_{max}}{1 + K_L C_e} \right] \tag{4}$$

where $q_e$ is the equilibrium adsorption (mg/g), $K_L$ is the constant of Langmuir, $C_e$ is the residual concentration of dye in water at equilibrium and $Q_{max}$ is the maximum adsorption of dye measured at equilibrium. The Hanes-Woolf linear form (Eq. (5)) of the Langmuir model was used to calculate the values of Langmuir model parameters from the respective curves (*Salah Ud Din et al., 2019*; *Imran et al., 2021*).

$$\frac{C_e}{q_e} = \left[ \frac{1}{K_L Q_{max}} + \frac{C_e}{Q_{max}} \right]. \tag{5}$$

According to Eq. (5), curves were made between $C_e$ *versus* $C_e/q_e$ to calculate Langmuir model parameters from the intercept $1/K_L Q_{max}$ and slope $1/Q_{max}$ of the curves (*Naeem et al., 2019*) for NiO-NPs, CuO-NPs and NiO/CuO-NC.

Another model used for equilibrium sorption was Freundlich model. Freundlich model gives adsorption of dyes on heterogeneous surfaces of an adsorbent expressed as;

$$q_e = K_F C_e^{1/n} \tag{6}$$

where $q_e$ is dye adsorption at equilibrium, $K_F$ is adsorption constant for Freundlich model, and 1/n is adsorption intensity of Freundlich model that shows the heterogeneity of adsorbents surface.

The linearized form Eq. (7) of Freundlich model was used to determine parameters of Freundlich model from the slope and intercept of the curve (*Imran et al., 2019*; *Naeem et al., 2019*).

$$\ln(q_e) = \left[ \ln(K_F) + \left( \frac{1}{n} \right) \ln(C_e) \right]. \tag{7}$$

## Adsorption kinetics

In order to determine the kinetics of the adsorption of RR-2 and OII onto the nanomaterials NiO-NPs, CuO-NPs and NiO/CuO-NC, optimum dose of the catalysts *i.e.*, 0.06 g 10mL$^{-1}$, and dyes concentration *i.e.*, 60 mg L$^{-1}$ and were worked at pH 7.5. Aliquots of the mixture

were taken after 15, 30, 60, and 120 min of shaking to determine the dyes absorbance at 540 nm and 485 nm, respectively. The linearized form of pseudo-first order (Eq. (8)), (*Gunasundari & Senthil Kumar, 2017*; *Shah et al., 2018*) and pseudo-second order (Eq. (9)) (*Matouq et al., 2015*; *Salah Ud Din et al., 2019*) were used to simulate the experimental kinetic adsorption of RR-2 and OII ono the surface of NiO-NPs, CuO-NPs and NiO/CuO-NC.

$$\log(q_e - q_t) = \log(q_e) - \left(\frac{k_1}{2.303}\right) \times t \tag{8}$$

where $q_e$ presents dyes adsorption at equilibrium; $q_t$ is dye adsorption with respect to time $t$ and $k_1$ israte constant of pseudo-first order. The slope ($k_1/2.033$) and intercept ($\log(q_e)$) of the curve between $\log(q_e - q_t)$ *versus* $t$ were used to calculate the values of model parameters (*Salah Ud Din et al., 2019*; *Imran et al., 2021*).

$$\frac{t}{q_t} = \frac{1}{k_2(qe)^2} + \frac{t}{q_e}. \tag{9}$$

Equation (9) shows the pseudo 2nd order kinetic model, where $k_2$ is pseudo-second-order rate constant. The slope $1/q_t$ and intercept $1/k_2(qe)^2$ of the curve between $t$ *versus* $t/q_e$ were used to calculate the values of pseudo 2nd order kinetic model parameters (*Iqbal et al., 2019*; *Naeem et al., 2019*).

# RESULTS AND DISCUSSIONS

## Characterization of NiO-NPs, CuO-NPs, and NiO/CuO-NC
### SEM analysis

Scanning electron microscopy (SEM) was used to reveal the surface morphology and particle size of the synthesized nanomaterials. The SEM results of CuO-NPs, NiO-NPs and NiO/CuO-NC have been shown in Fig. 1. ImageJ software was used to estimate particle size from SEM results. The SEM results reveal a variation in particles size and shape with change in material. The morphological study of CuO-NPs showed nearly uniform sized clusters of crystalline shaped particles having nanorods like structure (Fig. 1A). The mean particle size of CuO-NPs was found to be 123 nm while NiO-NPs exhibited irregular spherical morphological structures like snowflakes (Fig. 1B). The average particle size of NiO-NPs was recorded as 137 nm. The morphology of NiO/CuO-NC was found to be crystalline in shape and the average particle size of the nanocomposite was 73 nm. This smaller size of the nanocomposite may help to improve its potential for degradation of RR-5, RR-2, and OII dyes (Fig. 1C) from contaminated water as compared with NiO-NPs and CuO-NPs.

### XRD analysis

The X-ray diffraction pattern of CuO-NPs was recorded in the range of 20° to 60° (Fig. 2A). Peaks recorded at 32.65°, 35.01°, 38.50°, 48.60°, 53.62°, 58.72°, and 60.71° corresponded to the miller indices (hkl) values at (110), (002), (111), (202), (020), and (113), respectively. These peaks are well matched with JCPDS card number 45-0937 (*Mahmoud et al., 2020*). The average crystallite size of CuO-NPs crystals obtained with Debye Sherrer equation was found to be 14 nm.

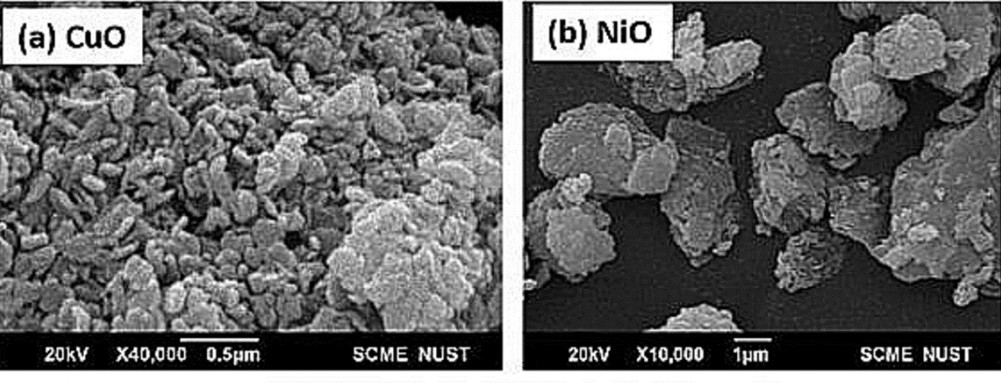

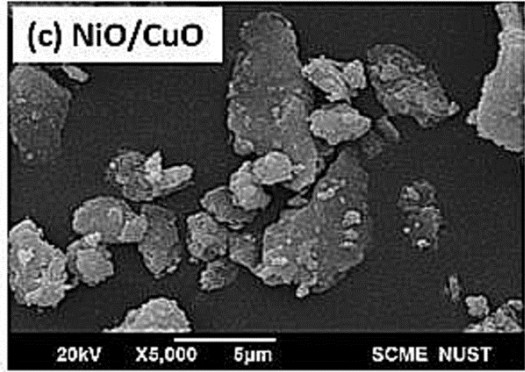

**Figure 1** **Scanning electron microscopic images of the synthesized nanomaterials.** The scanning electron microscopic images of (A) copper oxide nanoparticles (CuO-NPs), (B) nickel oxide nanoparticles (NiO-NPs), and (C) nanocomposite of NiO and CuO (NiO/CuO NC).

The XRD pattern of NiO-NPs shows agglomeration of crystallites. The diffraction peaks at 31.79°, 34.44°, 36.27°, 47.57°, and 56.63° closely resemble with crystalline planes (002), (111), (200), (020), and (202), respectively and the lattice parameters are in good agreement with JCPDS card number 47-1049 (*Srivastava & Srivastava, 2010*). Average particle size of NiO-NPs was 31 nm (Fig. 2B). Another study stated the 67nm crystallite size of chemically synthesized NiO-NPs at (111), (200), and (311) diffraction peaks (*Srivastava & Srivastava, 2010*).

The diffraction peaks of NiO/CuO-NC at 34.45° 42.67°, 62.34° and 32.59°, 37.50°, 44.73°, 48.18°, 53.50°, 54.63° match with hkl values [(111), (200), (220)] and [(111), (111), (112), (202), (020), (202)] which shows good agreement with JCPDS (card number 47-1049 and 45-0937) corresponding to NiO-NPs and CuO-NPs, respectively. The average crystallite size of nanocomposite was 13 nm (Fig. 2C).

### FTIR analysis

To unravel the functional groups present on the nanomaterials, FTIR spectra of these materials were taken at a wavenumber range from 4,000 $cm^{-1}$ to 300 $cm^{-1}$.

In FTIR spectrum of CuO-NPs, transmission at wavenumber 3,390.54 $cm^{-1}$ is attributed to O-H stretching mode (Fig. 3A and Table 1). The absorption peak at 1,631.40$cm^{-1}$ is

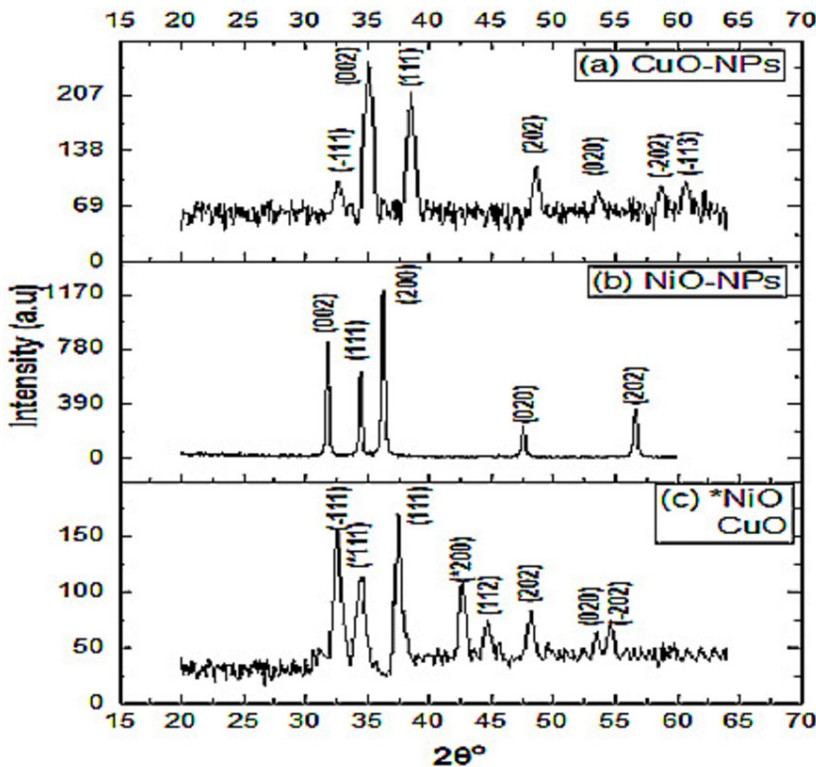

**Figure 2** **X-ray diffraction spectra of the nanomaterials.** X-ray diffraction spectra of (A) CuO-NPs, (B) NiO-NPs and (C) NiO/CuO-NC.

assigned to H−O−H molecule indicating the presence of water molecule by hydroxyl group (*Phiwdang et al., 2013*). The bands emerged at 1,123.49, 1,086.96, 985.02, and 730.98 cm$^{-1}$ are assigned to stretching vibrations of carbonyl group combined with copper atoms (*Dubal et al., 2010*). The sharp and noticeable bands at 595.75 and 520.5 cm$^{-1}$ confirmed the stretching vibration of Cu-O atom.

In the FTIR analysis of NiO-NPs, the broad bands in the range of 3,640.19 cm$^{-1}$ and 1,633.68 cm$^{-1}$ are linked to stretching mode of hydroxyl group O-H and H−O−H (Fig. 3B). It confirmed the presence of water molecule (*Ezhilarasi et al., 2016*). The weak bands at 1,145.54 cm$^{-1}$ and 1,047.02 cm$^{-1}$ are ascribed to the vibration and stretching peaks of carbonyl group (C=O). The sharp band at 519.11 cm$^{-1}$ is assigned to the presence of Ni (Ni-O-H) (*Anand et al., 2020*). The peak at wavenumber 457.29 cm$^{-1}$ corresponds to the Ni-O band.

The broad band in the region of 3,446.49 to 1,633.56 cm$^{-1}$ and 1,384.71 cm$^{-1}$ and 1,107.598 cm$^{-1}$ indicates the presence of O-H group and carbonyl group on the surface of NiO/CuO-NC (Fig. 3C). The bands at 617.15 cm$^{-1}$ and 410.16 cm$^{-1}$ are assigned to absorptive vibrations of Cu-O (*Shahsavani, Feizi & Dehno, 2016*) and Ni-O (*Barzinjy et al., 2020*), respectively.

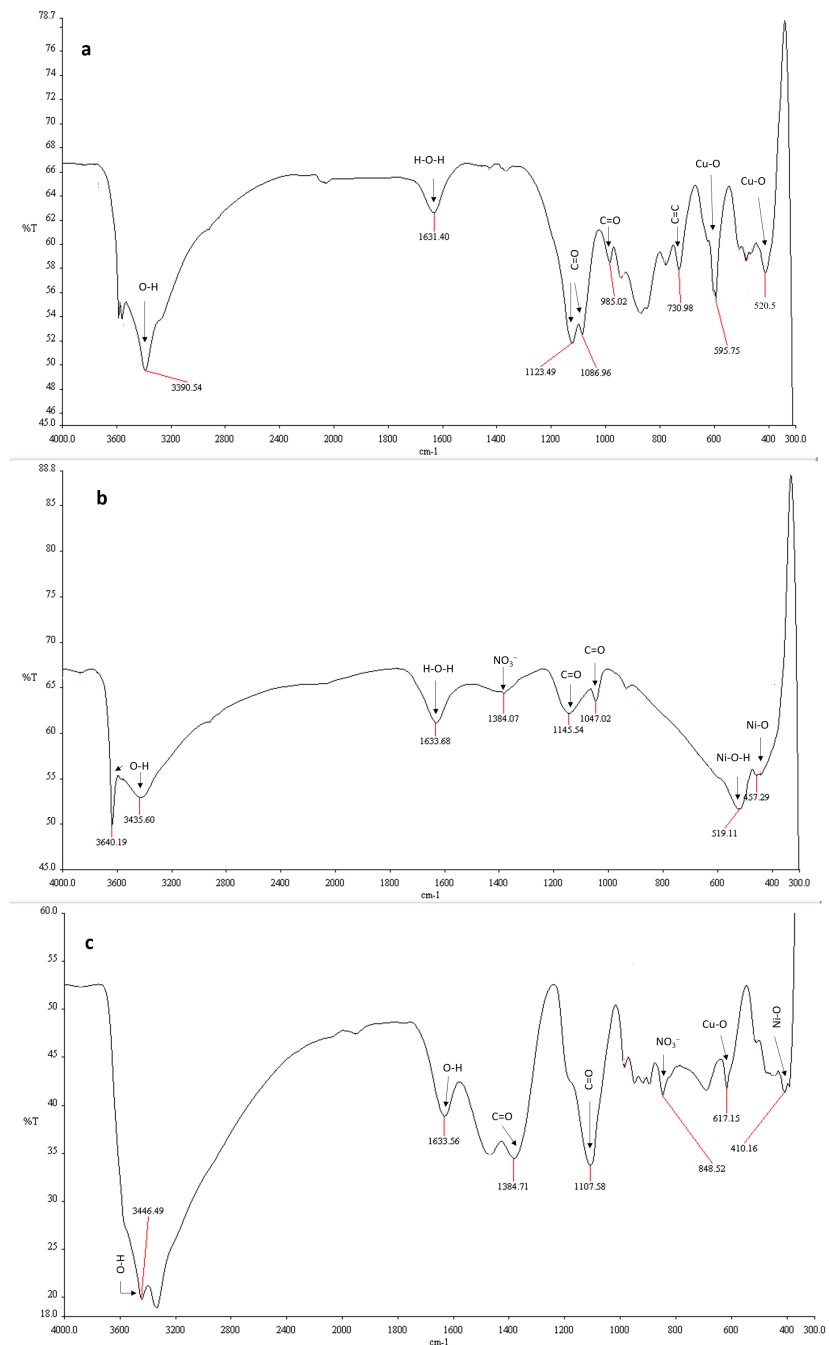

**Figure 3 FTIR spectra of the nanomaterials.** Fourier transform infrared spectra of (A) CuO-NPs, (B) NiO-NPs, and (C) NiO/CuO-NC.

## Factors affecting photocatalytic degradation

In this study, photocatalytic activity of p-p semiconductors was determined under various conditions. The comparison of photocatalytic activity of prepared nanocomposite with

bare nanoparticles towards the dyes degradation showed better photocatalytic activity under the same conditions.

## Effect of catalyst dose on photocatalysis of dyes

Catalyst dose is an important parameter to determine the photocatalytic efficiency of nanomaterials. The increment in % degradation was sharper at $0.02-0.06$ g 10 mL$^{-1}$ and slightly increased till equilibrium value 0.1 g 10 mL$^{-1}$, where 60 mg L$^{-1}$ dye concentration was chosen as optimal dye concentration. This is because the binding of all the dye molecules onto the surface of nanomaterials and formation of equilibrium between attached dye molecules on nanomaterials and in the solution (*Akar et al., 2009*).

Precisely, the degradation of RB-5 increased from 22.0 to 84.4%, that of RR-2 from 33.6 to 90.0%, and that of OII from 18.0 to 83.7% when dose of the catalyst NiO-NPs was increased from 0.02 to 0.1 g 10 mL$^{-1}$. For the same increase in the dose of the catalyst CuO-NPs, the degradation of RB-5 merely increased from 7.5 to 34.3%, that of RR-2 from 13.13 to 49.3%, and that of OII from 15.0 to 44.0%. However, for the same increase in the dose of the catalyst NiO/CuO-NC, the degradation of RB-5 increased from 69.0 to 92.8%, that of RR-2 from 39.0 to 92.0%, and that of OII from 76.2 to 96.2%. Results showed the high degradation efficiency of dyes at high catalyst dose (0.1 g 10mL$^{-1}$) under irradiation (420 nm) of 2 h (Fig. 4). Moreover, NiO/CuO-NC showed higher degradation rate than CuO-NPs but almost equivalent to NiO-NPs at high catalyst dose that is consistent with the results of previous study (*Balasubramani, Sivarajasekar & Naushad, 2020*).

The NiO/CuO-NC exhibits excellent potential to efficiently degrade the dyes with increase in its dose. According to characterization results (Fig. 1C), high catalyst dose of NiO/CuO-NC increases the number of active sites and driving accumulation forces on the surface of nanomaterial, which further bind high amount of dye molecules on its surface and vice versa (*Nautiyal, Subramanian & Dastidar, 2016*). The higher degradation trend at high dose was attributed to availability of small particle size and more capacity to absorb the dye molecules (*Vargas-Reus et al., 2012*). Additionally, (i) the superoxide and hydroxyl radicals in nanocomposite may also increase the rate of degradation of dyes, and (ii) photodegradation activity is also based on electron donor and acceptor where surface of nanomaterials acts as a donor to accept the electrons of dye molecules on its surface (*Sun, Xiao & Wu, 2019*). A similar observation was reported by *Moradi, Haghighi & Allahyari (2017)* for degradation of Acid Orange (AO7) that increased from 93–100% at higher dose of Ag/ZnO-NPs (0.75–2 g L$^{-1}$).

## Effect of initial dye concentration

The impact of dyes initial concentration on their degradation at a fixed dose of the catalyst *i.e.*, 0.06 g 10 mL$^{-1}$, pH 7.5, and irradiation of 420 nm for 2 h was studied by varying the levels of RB-5, RR-2, and OII dyes from 20–100 mg L$^{-1}$. With the increasing initial concentration of dyes, the degradation of RB-5 decreased from 99.5 to 44.6%, 71.8 to 41.6%, and 100 to 85.7% in response to NiO-NPs, CuO-NPs, and NiO/CuO-NC respectively (Fig. 5). Similarly, the degradation of RR-2 decreased from 86.1 to 45.9%, 42.0 to 18.4%, and 86.8% to 78.8% in response to NiO-NPs, CuO-NPs, and NiO/CuO-NC

Ahsan et al. (2022), *PeerJ*, DOI 10.7717/peerj.14358

**Table 1   FTIR spectral bands of CuO-NPs, NiO-NPs, and NiO/CuO-NPs.**

| Wavenumber (cm$^{-1}$)-Cu-O | Vibration assignment | Wavenumber (cm$^{-1}$)- Ni-O | Vibration assignment | Wavenumber (cm$^{-1}$)-NiO/CuO | Vibration assignment |
|---|---|---|---|---|---|
| 3,390.54 cm$^{-1}$ | O-H | 3640.19 cm$^{-1}$, 1633.68 cm$^{-1}$ | O-H, H−O−H | 3446.49 to 1, 633.56 cm$^{-1}$ | O-H, H−O−H |
| 1,631.40 cm$^{-1}$ | H−O−H | 1145.54 cm$^{-1}$, 1047.02 cm$^{-1}$ | Carbonyl group | 1,384.71 cm$^{-1}$, 1107.598 cm$^{-1}$ | Carbonate group |
| 1,123.49, 1086.96, 985.02, and 730.98 cm$^{-1}$ | Carbonyl group | 519.11 cm$^{-1}$ | Ni-O-H | 617.15 cm$^{-1}$ | Cu-O |
| 595.75 and 520.5 cm$^{-1}$ | Cu-O | 457.29 cm$^{-1}$ | Ni-O | 410.16 cm$^{-1}$ | Ni-O |
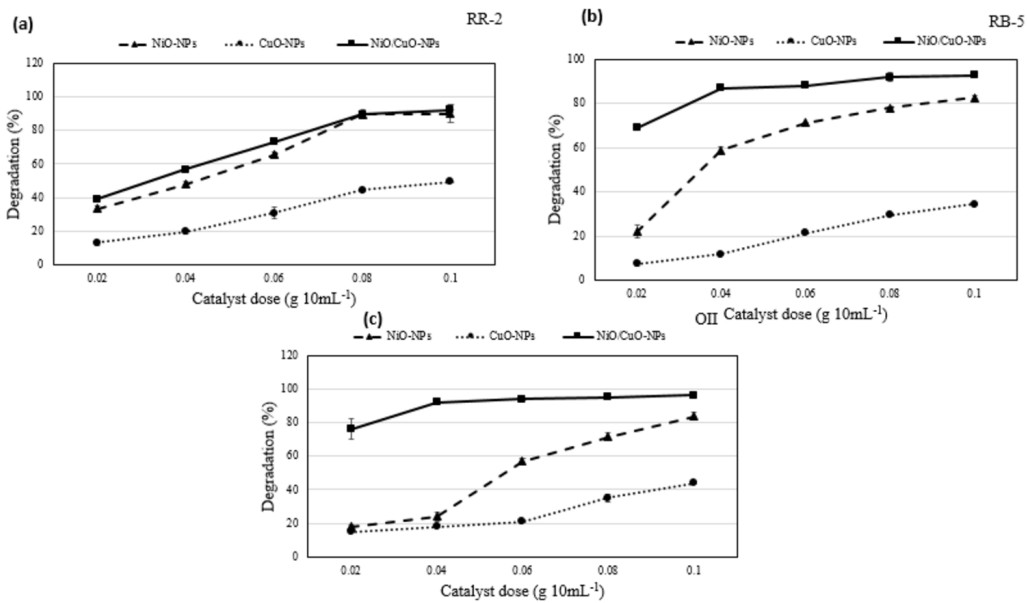

**Figure 4 Degradation potential of nanomaterials.** Effect of catalyst dose on photocatalytic degradation potential of nanomaterials against (A) Reactive-Red 2, (B) Reactive Black-5, and (C) Orange II.

respectively. Moreover, the degradation of OII when its initial concentration was increased from 20 to 100 mg L$^{-1}$, changed from 92.0 to 73.5%, 51.1 to 52.4%, and 87.4% to 80.5% in response to NiO-NPs, CuO-NPs, and NiO/CuO-NC respectively.

Overall, the percentage degradation decreased with increasing dye concentration. This could be correlated to limited number of active sites on the nanomaterials which become saturated with higher concentration (*Hassan & Carr, 2021*). However, the nanocomposite showed higher dye degradation potential than NiO-NPs and CuO-NPs with increase in initial concentration of dyes (20 to 100 mg L$^{-1}$). Literature proved that integration of both nanomaterials induced extension in excitation wavelength towards visible light irradiation (520 nm) rather than individual nanomaterial (*Zhu et al., 2018*).

Dye degradation by the CuO-NPs was not significant. Literature shows 60% degradation of Methyl Orange (MO) within 1 h of UV irradiation using CuO nanofibers. Nevertheless, low degradation of dyes was observed because of the scarcity of OH$^{.}$ radicals on nanomaterial (*Almehizia et al., 2022*). It is evident that dye degradation is correlated with the formation of OH$^{.}$ and O$_2^-$ and their reaction with dye molecules. The less degradation at high concentration of dyes may be due to (i) significant irradiation of visible light on dye molecules which reduces the catalytic reaction efficiency, (ii) increase in dye amount or intense colour that decreases the path length of photons entering into the solution, and (iii) coating of more dyes ions leading to less number of OH$^{.}$ and O$_2^-$ formation thereby preventing effective reaction of OH$^{.}$ andelectron holes with dye molecules to degrade the dyes (*Hegazey et al., 2020*). NiO/CuO-NC showed higher degradation than NiO-NPs and CuO-NPs even at higher initial concentration of dyes due former's smaller particle size,

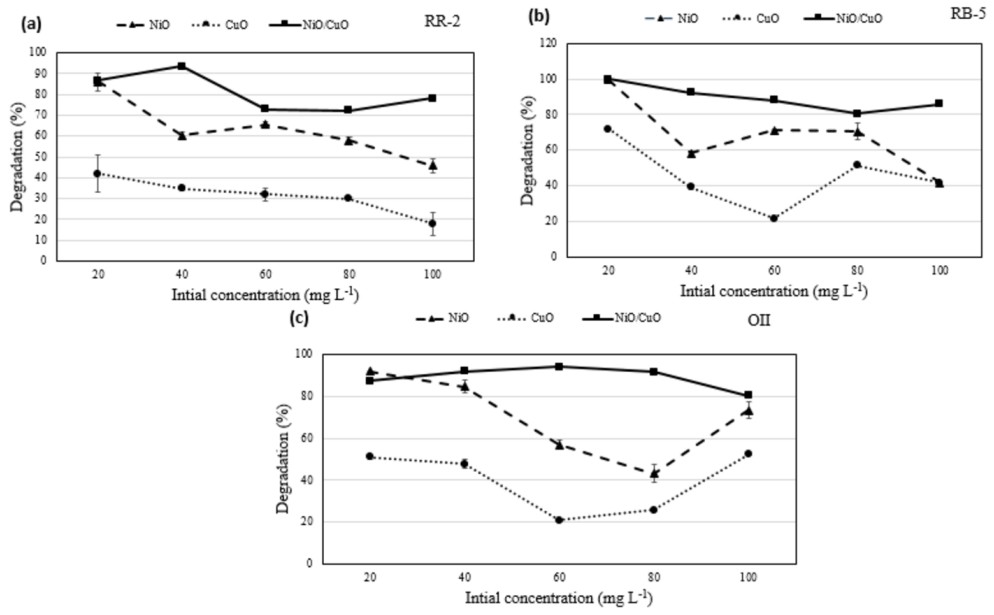

**Figure 5   Effect of initial concentration of dyes on photodegradation.** Effect of initial dyes concentration of (A) Reactive-Red 2, (B) Reactive Black-5, (C) Orange II, on photocatalytic degradation potential of nanomaterials.

and more active sites to attract the additional electrons from dye molecules than individual nanomaterials.

The results of this study are in accordance with another research in which authors studied the effect of initial dye concentration (8–12 mg L$^{-1}$) on degradation efficiency of nanomaterials and found the maximum degradation at 10 mg L$^{-1}$ (*Anju Chanu et al., 2019*). Similarly, 100% of RB-5 was degraded by 0.5 mg L$^{-1}$ of Fe/Pd composite when the former's concentration was 40 mg L$^{-1}$ (*Samiee, Goharshadi & Nancarrow, 2016*). However, the degradation potential of the Fe/Pd decreased to 95.5%, 94.3%, and 90.8% when the initial concentration of RB-5 was increased to 60, 80, and 100 mg L$^{-1}$ respectively.

## Modelling adsorption isotherm

The experimental equilibrium adsorption data for RR-2 and OII dyes by NiO-NPs, CuO-NPs and NiO/CuO-NC was simulated with Langmuir and Freundlich isotherm models. Correlation behaviour of both models with the experimental data has been shown in Figs. 6A–6D. For complete understanding of RR-2 and OII dyes adsorption, $q_e$ *vs* $C_e$ for NiO-NPs, CuO-NPs and NiO/CuO-NC has been presented in Figs. 6E–6F. The parameters including the values of correlation coefficient for Langmuir and Freundlich models for the removal of RR-2 and OII by NiO-NPs, CuO-NPs and NiO/CuO-NC adsorbents have been presented in Table 2. The $R^2$ values indicate that Freundlich isotherm model well depicted the adsorption of RR-2 on NiO-NPs, CuO-NPs and NiO/CuO-NC and OII adsorption onto the surface of NiO-NPs, and CuO-NPs. While OII adsorption onto NiO/CuO-NC was well-described by Langmuir model. The *n* values for RR-2 adsorption with all the three

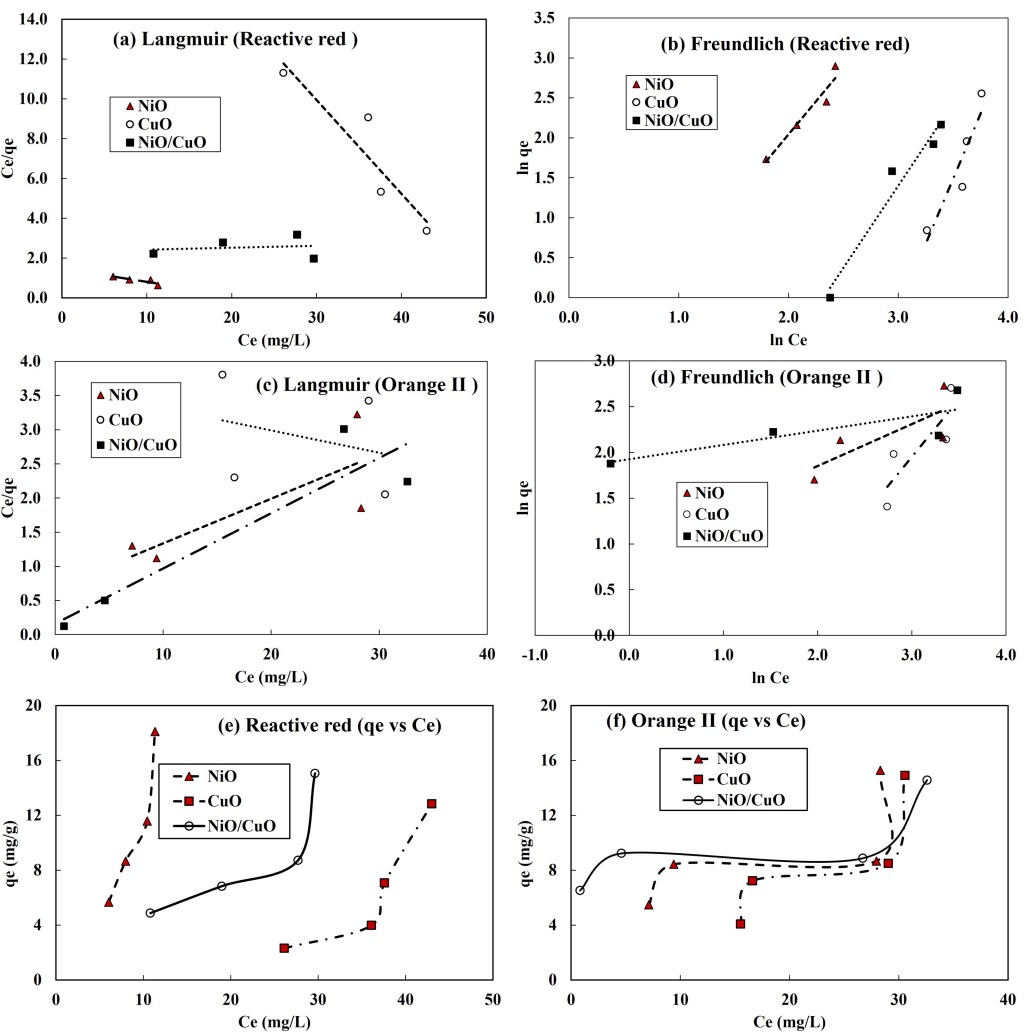

**Figure 6    Isotherm model fitting to adsorption of dyes onto nanomaterials.** Langmuir model (A, C), Freundlich model results (B, D), and plots of equilibrium adsorption (qe, mg/g) against residual dye concentration ($C_e$, mg/L) at equilibrium (E, F), for RR-2 dye and OII dye, respectively.

adsorbents are less than one while n>1 for the adsorption of OII onto the surface of all the three adsorbents (NiO-NPs, CuO-NPs and NiO/CuO-NC). The maximum value of *n* *i.e.*, 6.41 was found for the adsorption of OII by nanocomposite (NiO/CuO-NC). Table 2 shows the values of the Langmuir and Freundlich models, which suggest that the best fitted Freundlich model ($R^2 = 0.93$, 0.87, and 0.96) *versus* the Langmuir model (0.76, 0.86, and 0.024) for the removal of RR-2 by NiO-NPs, CuO-NPs and NiO/CuO-NC, respectively.

The $R^2$ is a correlation coefficient for the experimental and numerical adsorption data and its value varies with the adsorbent materials, contaminant, and other experimental conditions. The reason behind the difference in $R^2$ for different models is that each model has several assumptions for its application depending on the adsorbent type and contaminant type under given conditions. In addition to the adsorbent and adsorbate

**Table 2  Adsorption isotherm constants for adsorption of RR-2 and OII dyes onto various nanomaterials.**

| Model | | Langmuir | | | Freundlich | | |
|---|---|---|---|---|---|---|---|
| Parameters | Adsorbent | $Q_{max}$ | $K_L$ | $R^2$ | $n$ | $K_F$ | $R^2$ |
| Reactive Red-2 | NiO-NPs | 15.04 | 0.045 | 0.76 | 0.6 | 3.5 | 0.93 |
| | CuO-NPs | 2.12 | 0.02 | 0.86 | 0.31 | 21807.3 | 0.87 |
| | NiO/CuO-NC | 103.09 | 0.004 | 0.024 | 0.49 | 116.7 | 0.96 |
| Orange II | NiO-NPs | 15.31 | 0.095 | 0.62 | 2.19 | 2.6 | 0.61 |
| | CuO-NPs | 30.67 | 0.009 | 0.093 | 0.8 | 6.2 | 0.71 |
| | NiO/CuO-NC | 12.36 | 0.509 | 0.86 | 6.41 | 6.9 | 0.66 |

nature, another factor that might affect the $R^2$ values could be pH and temperature (*Tran et al., 2017*). Environmental parameters particularly solution pH and temperature must be controlled over the entire experiment. In the current study pH and temperature have been studied invariably and their impact on the adsorption for selecting optimum level was not evaluated.

## Modelling adsorption kinetics

Pseudo-first order (PFO) and pseudo-second order (PSO) kinetic models were tested to obtain kinetic adsorption capacity of nanomaterials at different time, at constant pH and dose (0.06 g 10 mL$^{-1}$). The suitability of the models was evaluated based on the value of correlation coefficient ($R^2$) (*Shah et al., 2018*; *Imran et al., 2019*). The correlation behaviour for the adsorption of RR-2 and OII dyes onto three adsorbents have been shown in Figs. 7A–7D while adsorption kinetics ($q_t$ *vs* t) of RR-2 and OII dyes onto NiO-NPs, CuO-NPs and NiO/CuO-NC have been presented in Figs. 7E–7F). Table 3 presents the values of kinetic models' parameters. The modelling results revealed that $R^2$ values for RR-2 adsorption onto NiO-NPs were well explained by PSO while RR-2 adsorption onto CuO-NPs and NiO/CuO-NC was equally described with PFO and PSO ($R^2 \geq 0.97$). Moreover, the adsorption of OII dye onto NiO-NPs and NiO/CuO-NC best fitted with PSO model while its adsorption onto CuO surface was equally represented by PSO and PFO kinetic models as per values of $R^2$.

## Comparison of dye-degradation efficiency of nanomaterials of this study with literature

The comparison study of nanomaterials used in this research with previous literatures is vital to compare the degradation efficiency and cost effectiveness of proposed nanomaterials (Table 4). Present study highlights the degradation efficiency of RB-5, RR-2, OII dyes using NiO/CuO-NC and NiO-NPs was (100%, 87%, and 87%), and (99%, 86%, and 92%), respectively. It was observed that the synthesized materials are more efficient to degrade the dyes in less time than cerium nanoparticles, Ce-NPs (*Muthuvel et al., 2021*), iron nanoparticles, Fe-NPs (*Xu et al., 2012*), Titanium oxide, TiO$_2$-NPs (*Ezzatahmadi et al., 2018*), nickle doped titanium oxide nanocomposite, Ni/TiO$_2$-NC (*Kamaludin et al., 2019*) and stannic oxide, SnO$_2$ (*Khan et al., 2020*). Other nanomaterials were also efficient to

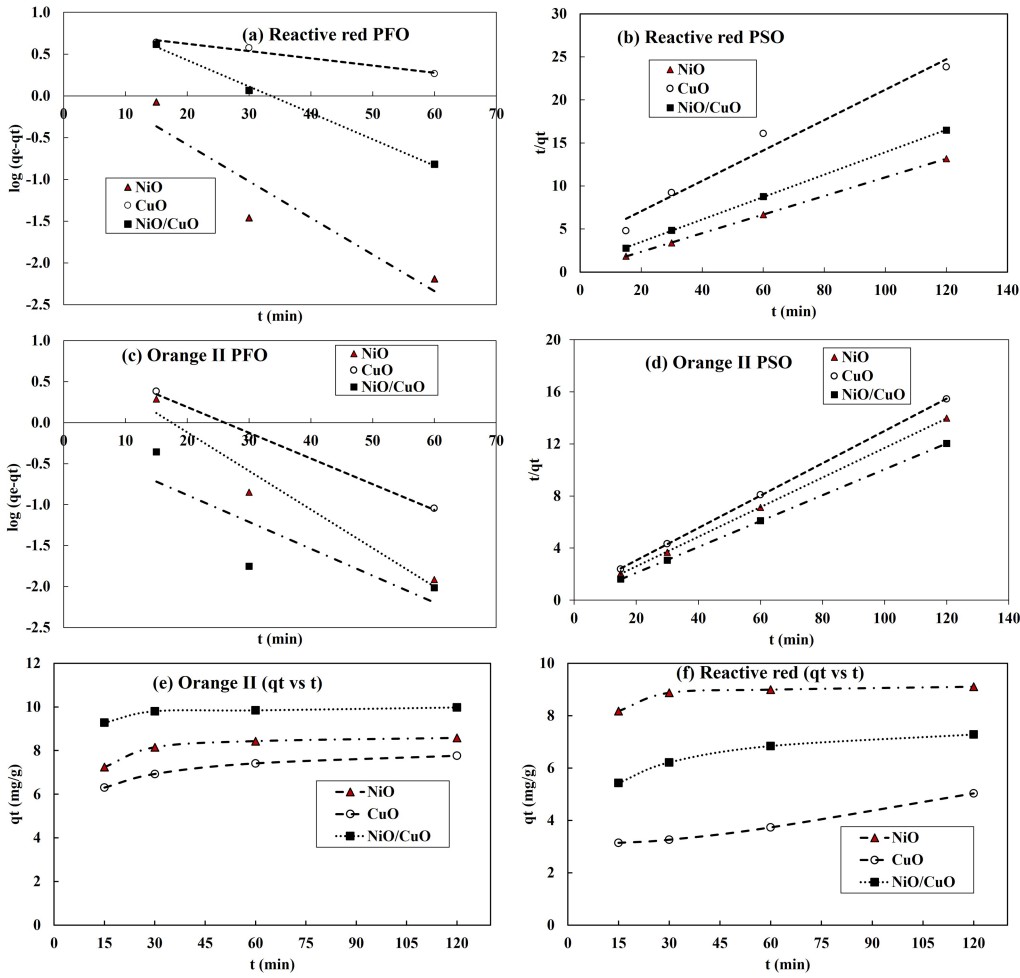

**Figure 7  Kinetics adsorption parameters of Reactive Red-2 and Orange-II.** Pseudo-first order and pseudo-second order results for RR-2 dye (A, B) and Orange II (C, D) respectively, and kinetics of dye adsorption of RR-2 (E), and Orange II (F).

degrade the dyes but the efficiency of NiO/CuO-NC for dye degradation makes it attractive due to their cost-effectiveness and high degradation ability.

## CONCLUSIONS

The NiO-NPs, CuO-NPs, and NiO/CuO-NC were synthesized successfully and used to monitor the photocatalytic reaction over short time (2 h) of visible light irradiation (420 nm) which was considerably less time duration for the dyes degradation as compared with literature. Photodegradation potential of these nanomaterials was evident with SEM, XRD, and FTIR analysis with good surface area, presence of functional groups and small particle size. Under optimized conditions, the NiO/CuO-NC exhibits stronger photocatalytic activity of 92, 93, and 96% against RR-2, RB-5, and OII dyes respectively, than other nanomaterials. The dyes degradation pattern throughout the experiment

**Table 3  Kinetics adsorption parameters *e.g.*, dye concentration (60 mgL$^{-1}$), dose (0.06 g 10 mL$^{-1}$ and pH (8) for the adsorption of RR-2 and OII dyes onto various nanomaterials.**

| Model | | PFO | | | PSO | | |
|---|---|---|---|---|---|---|---|
| Parameters | Adsorbent | $k_1$ | qe.cal | $R^2$ | qe.cal | $k_2$ | $R^2$ |
| Reactive Red-2 | NiO-NPs | 0.101 | 1.34 | 0.87 | 9.23 | 0.068 | 1.000 |
| | CuO-NPs | 0.020 | 2.21 | 0.97 | 5.67 | 0.009 | 0.970 |
| | NiO/CuO-NC | 0.073 | 2.88 | 0.99 | 7.67 | 0.019 | 0.990 |
| Orange II | NiO-NPs | 0.108 | 2.28 | 0.95 | 8.79 | 0.042 | 0.999 |
| | CuO-NPs | 0.072 | 2.26 | 0.99 | 8.06 | 0.027 | 0.999 |
| | NiO/CuO-NC | 0.076 | 1.25 | 0.71 | 10.10 | 0.087 | 0.999 |

**Table 4  Comparison of dye-degradation efficiency of nanomaterials of this study with literature.**

| No | Nanomaterials | Degradation | | | Reference |
|---|---|---|---|---|---|
| | | RB-5 | RR-2 | OII | |
| 1 | NiO/CuO-NC | 100% | 87% | 87% | This study |
| 2 | NiO-NPs | 99% | 86% | 92% | This study |
| 3 | Ce-NPs | | 78% | | Lucas et al., 2017 |
| 4 | Fe-NPs | | | 71% | *Xu et al. (2012)* |
| 5 | TiO$_2$-NPs | 58% | | | *Ezzatahmadi et al. (2018)* |
| 6 | NP/NiO-NC | | 93% | | *Hafdi et al. (2020)* |
| 7 | Cu$_2$O-NPs | | 90% | | *Halbus et al. (2022)* |
| 8 | Ni/TiO$_2$-NC | 75% | | | *Kamaludin et al. (2019)* |
| 9 | CuFe$_2$O$_4$-NPs | 98% | | | *Kodasma et al. (2020)* |
| 10 | TiO$_2$-NPs | | 80% | | *Badmus et al. (2021)* |
| 11 | SnO$_2$-NPs | | | 65% | *Khan et al. (2020)* |
| 12 | ZnO-NPs | 78% | 76% | | *Siddique et al. (2021)* |

was: NiO/CuO-NC >NiO-NPs >CuO-NPs. The adsorption modelling of isotherm and kinetics experimental data showed best fit with the Freundlich model ($R^2 = 0.96$) and the pseudo second order kinetic model ($R^2 \geq 0.97$), respectively. The improvement in dyes degradation by the nanocomposite is attributed to the combined incorporation of CuO-NPs and NiO-NPs as catalyst with significant better adsorption, photocatalytic and electron transfer activity of nanocomposite. The prepared nanocomposite with high surface area could be introduced as a suitable nanomaterial in waste water remediation.

### Funding

This research was funded by Researchers Supporting Project number RSP-2021/123 King Saud University, Riyadh, Saudi Arabia. Researchers Supporting Project number (RSP-2021/123) King Saud University, Riyadh, Saudi Arabia are acknowledged. The funders had no role in study design, data collection and analysis, decision to publish, or preparation of the manuscript.

## Grant Disclosures

The following grant information was disclosed by the authors:
Researchers Supporting Project:  RSP-2021/123.
King Saud University, Riyadh, Saudi Arabia.

## Competing Interests

Muhammad H. Siddique is an Academic Editor for PeerJ. However, he was not involved in handling this manuscript as an editor at any stage. All the other authors declare no competing interest.

## Author Contributions

- Hajra Ahsan conceived and designed the experiments, performed the experiments, prepared figures and/or tables, authored or reviewed drafts of the article, and approved the final draft.
- Muhammad Shahid conceived and designed the experiments, authored or reviewed drafts of the article, and approved the final draft.
- Muhammad Imran analyzed the data, prepared figures and/or tables, and approved the final draft.
- Faisal Mahmood performed the experiments, authored or reviewed drafts of the article, and approved the final draft.
- Muhammad Hussnain Siddique performed the experiments, authored or reviewed drafts of the article, and approved the final draft.
- Hayssam M. Ali analyzed the data, authored or reviewed drafts of the article, provided funds for analysis and publication, and approved the final draft.
- Muhammad B.K. Niazi performed the experiments, prepared figures and/or tables, and approved the final draft.
- Sabir Hussain conceived and designed the experiments, authored or reviewed drafts of the article, and approved the final draft.
- Muhammad Shahbaz conceived and designed the experiments, authored or reviewed drafts of the article, and approved the final draft.
- Mudassar Ayyub performed the experiments, prepared figures and/or tables, and approved the final draft.
- Tanvir Shahzad conceived and designed the experiments, analyzed the data, prepared figures and/or tables, and approved the final draft.

## Data Availability

The raw data is available as a Supplementary File.

## Supplemental Information

Supplemental information for this article can be found online at http://dx.doi.org/10.7717/peerj.14358#supplemental-information.

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
