# Peer review of "Photocatalysis and adsorption kinetics of azo dyes by nanoparticles of nickel oxide and copper oxide and their nanocomposite in an aqueous medium"

_PeerJ, doi:10.7717/peerj.14358_

## Round 0.1 · original submission · Major Revisions

Please revise the manuscript according to the reviewers' comments.

Reviewer 1 ·

Basic reporting

This manuscript reported the preparation of nickel oxide (NiO-NPs), copper oxide (CuO-NPs) and the nanocomposite (NiO/CuO-NC) for degradation of dyes. The work is well designed. I recommended the acceptance of this article for publication after addressing the following issues.
1. The irradiation light energy intensity should be given.
2. In the section of introduction, the advances of photocatalysts should be improved. eg: Advanced Powder Materials, 2023, 2, 100073; Journal of Materials Science and Technology, 2022, 123, 177-190; Chem. Eng. J., 2022, 428, 131158; J. Colloid Interface Sci., 2022, 624, 219-232; Separation and Purification Technology, 2022, doi: 10.1016/j.seppur.2022.121892; Inorganic Chemistry Frontiers, 2022, 9, 2479-2497; J. Colloid Interface Sci., 2022, 619, 307-321; Chem. Eng. J., 2022, 429, 132519;
3. Cycling test should be performed to check its stability.

Experimental design

It is well designed

Validity of the findings

no comment

Additional comments

no comment

Reviewer 2 ·

Basic reporting

1. Figures quality must be improved.
2. Experimental results are needed to know surface area of the samples.

Experimental design

no comment

Validity of the findings

1. According to the given experimental condition, and observations (Lines 133 – 136, 140 and 141, and 148-150), the synthesized materials may be metal hydroxides. Please check the color of nickel and copper hydroxides.
2. Line 141 is confusing. How does the appearance of greenish grey color indicate morphology?
3. Line 146 – Metal salts (nickel and copper) will be dissolved in water. It may not be a suspension.
4. Line 48 - Please provide BET results for the samples.
5. Line 157 – Please check the wavelength value.
6. Line 159 – Check 2theta degree range
7. Line 161 – Please cite source reference.
8. Line 164 -  is the wavelength of X-ray.
9. Please check line – 91 (“…of CuO-NPs justify their applications in photocatalytic activity of NiO-NPs for dye..”)
10. Line 166 – Solid samples can be directly examined through ATR mode FTIR.
11. Line 167 - “..4000 cm-1 to 300 cm-1 wavelengths” – It should be wavenumber.
12. Lines 258 and 259 – the number of 2theta values and crystal plane values are not the same. Please check.
13. SEM results and their interpretation must be improved.
14. What is the source of carbonate peaks in FTIR spectra?
15. Line 291 – What is Ni-O-H? Is it nickel hydroxide?
16. Photocatalytic mechanism of NiO/CuO composite should be clearly explained.
17. Photocatalytic and adsorption performances should be compared with other reports in the table form.
18. Line 181 - Please provide lamp details.

Reviewer 3 ·

Basic reporting

Your most important issue is the interpretation of SEM images. Based on the image provided in the manuscript, it is hard to say that NiO-NPs are homogenous. There are the aggregation of NPs in all three images in figure 1.

Please add details of the mechanism of the co-precipitation method to help the readers of this journal to understand the process of NiO-NPs, CuO-NPs, and NiO/CuO-NC formation.

Please provide a hypothesis why the nanocomposite showed higher dye degradation potential than NiO-NPs and CuO-NPs with the increasing initial concentration of dyes.

Experimental design

Your method description needs to have more details and information to replicate. For example, the stir plate speed for the preparation of NiO-NPs (line 131) and how precipitates are washed (line 134). This comment applies to the preparation of CuO-NPs as well.

Additionally, I suggest that you provide the detail on how NiO-NPs were separated and shaped into fine powder.

Validity of the findings

Thank you for providing the raw data package. Please specify what software was used to measure the size of NPs prepared in this study using SEM images.

Additionally, there are negative values in the raw data for NPs size. Please provide an explanation of these negative values.

Reviewer 4 ·

Basic reporting

This article is fascinating work, and I am confident it would appeal to readers in the field. I recommend that the authors address the comments mentioned here and revise so that the quality of the manuscript is improved.

Minor comments:

1) "Azo dyes are obnoxious" - This is a vague and ambiguous statement as there are a lot of positive applications as well. The authors can remove this and say, "Azo dyes are organic pollutants present in..."

2) Please perform thorough grammar proofreading for the entire manuscript to get rid of errors.

3) The 3 dyes mentioned, are they anionic or cationic? Can the authors elaborate on the charge properties of these dyes and their implications in this manuscript? A paragraph in the manuscript should suffice.

4) The authors need to scientifically discuss the meaning of R2 values from the isotherm models.

Results need to be discussed scientifically and rationally in detail. For example: How do they explain the low r2 values in tables 2 and 3? 0.024, 0.09, 0.62, etc.

R2 values below 0.8 need to be explained and justified in the manuscript, and the rationale behind using the values.

Experimental design

5) For the spectroscopy studies, what cuvettes did the authors use? Kindly mention the specifications, brand, and manufacturer.

6) What magnetic stirrer was used for synthesis? What RPM?

7) For photocatalysis, what lamp setup was used? Lamp type? Intensity? Wavelengths? - These details are very critical to this study.

Authors are requested to check the manuscript for any other additional missing details in the manuscript wrt materials used and make sure they are taken care of.

Validity of the findings

5) For Langmuir and Freundlich isotherms, the authors are advised to read this paper (Mistakes and inconsistencies regarding adsorption of contaminants from aqueous solutions: A critical review) and make the necessary changes. This paper is highly relevant to this research and could help improve the quality of the findings significantly.

Link: https://www.sciencedirect.com/science/article/abs/pii/S0043135417302695

Additional comments

The authors are highly advised to provide a graphical abstract of their work as it would help readers get the overall idea of the article quickly and help readership and improve the impact. This can be considered a major recommendation. A graphical abstract would significantly help improve the impact of the authors' work.

---

## Round 0.2 · Minor Revisions

Please revise the manuscript according to reviewer 4's comments.

Reviewer 1 ·

Basic reporting

The revision is qualified for publication

Experimental design

Good

Validity of the findings

Good

Additional comments

No

Reviewer 2 ·

Basic reporting

Fine

Experimental design

Fine

Validity of the findings

Fine

Additional comments

Accept

Reviewer 3 ·

Basic reporting

The author has addressed my comments. More details of the mechanism of the co-precipitation method have been added to help the readers of this journal to understand the process of NiO-NPs, CuO-NPs, and NiO/CuO-NC formation. The hypothesis has been provided to explain why the nanocomposite showed higher dye degradation potential than NiO-NPs and CuO-NPs with the increase in the initial concentration of dyes.

Experimental design

More details have been added about the method description including the stir plate speed for the preparation of NiO-NPs and how precipitates are washed.

Validity of the findings

The author has addressed my comments about NP size using ImageJ.

Reviewer 4 ·

Basic reporting

The authors have addressed most of the comments. However, one significant comment needs to be discussed in the manuscript. The isotherms are flawed and need to be corrected. This is a major correction.

5) For Langmuir and Freundlich isotherms, the authors are advised to read this paper (Mistakes and inconsistencies regarding adsorption of contaminants from aqueous solutions: A critical review) and make the necessary changes. This paper is highly relevant to this research and could help improve the quality of the findings.

Although the authors have said that "The suggested article has been read for the current study and relevant future work. " and also have cited this (They have added it to the reference section and have linked it in line 211 without making any changes to their manuscript), they have NOT done any corrections to their manuscript in this aspect which is very critical. Please make the required corrections to the incorrect isotherms, and only then can this manuscript be published. Read the recommended review paper first, understand your paper's issues concerning the isotherms, and then incorporate the changes instead of just citing it.

I am NOT associated with the review paper I recommended to you, so I am not looking for any citations. I am looking for transparent corrections on the flaws of your isotherms before resubmission.

Experimental design

I am satisfied by the response of the authors.

Validity of the findings

Isotherms need to be corrected, and comments regarding that need to be addressed.
Figures 6 and 7 needs to be corrected with new isotherms / experiments.

Additional comments

Please make the changes with regards to the isotherms and provide your detailed response on this. Since the isotherms are misleading in some aspects, corrections need to be made in the main manuscript.

---

## Round 0.3 · accepted · Accept

The manuscript now is acceptable.